# Effect of acupuncture on patients with major psychiatric disorder and related symptoms caused by earthquake exposure: Protocol for a scoping review of clinical studies

**Hui-Ju Kwon**[1]ᵒ, **Jungtae Leem**[2]ᵒ, **Da-Woon Kim**[3], **Chan-Young Kwon**[4], **Sang-Ho Kim**[3]*

1 College of Korean Medicine, Daegu Haany University, Gyeongsan-si, Gyeongsangbuk-do, Republic of Korea, 2 College of Korean Medicine, Wonkwang University, Iksan, Republic of Korea, 3 Department of Neuropsychiatry of Korean Medicine, Pohang Korean Medicine Hospital Affiliated to Daegu Haany University, Pohang-si, Gyeongsangbuk-do, Republic of Korea, 4 Department of Oriental Neuropsychiatry, Dong-Eui University College of Korean Medicine, Busan, Korea

ᵒ These authors contributed equally to this work.
* omed22@naver.com

**Data Availability Statement:** No datasets were generated or analysed during the current study. All

## Abstract

Earthquakes have the greatest destructive effect among all natural disasters. Posttraumatic stress disorder (PTSD), major depressive disorder (MDD), and anxiety disorder (AD) are major psychiatric disorders (MPD) that can be triggered by exposure to earthquakes. Conventional treatments such as pharmacological treatments have several limitations. Acupuncture therapy as a complementary integrative medicine may be an effective alternative treatment for these limitations. This study aimed to identify the status of the clinical evidence regarding acupuncture therapy for earthquake survivors with MPD. We will follow the scoping review process as previously described. The study question is as follows: "Which types of clinical research designs, study types, study durations, adverse events, and clinical outcomes have been reported regarding acupuncture therapy for MPD in earthquake survivors?" Medline, Excerpta Medica dataBASE, Cochrane Central Register of Controlled Trials, Web of Science, Scopus, Allied and Complementary Medicine Database, Cumulative Index to Nursing and Allied Health Literature, PsycArticles databases, and Chinese, Korean, and Japanese databases will be comprehensively searched electronically from their inception to November 2022. Data from the included studies will be collected and descriptively analyzed in relation to our research question. We will collate, synthesize, and summarize the extracted data according to the analytical framework of a scoping review. The protocol will conform with the Preferred Reporting Items for Systematic Reviews and Meta-Analyses Extensions of Scoping Reviews to ensure the clarity and completeness of our reporting in the whole phase of the scoping review (Protocol registration: https://osf.io/wfru7/). The findings of this scoping review will provide fundamental data that will help researchers identify appropriate research questions and design further studies on the use of acupuncture for MPD management in earthquake survivors. These

relevant data from this study will be made available upon study completion.

**Funding:** This work was supported by the National Research Foundation of Korea (NRF) grant funded by the Korean government (MSIT) (No. 2021R1F1A105928211). SHK has received this fund. The funding source had no input in the interpretation or publication of the study results. The funders had no role in study design, data collection and analysis, decision to publish, or preparation of the manuscript. ※ MSIT: Ministry of Science and ICT.

**Competing interests:** The authors have declared that no competing interests exist.

results will be helpful for developing disaster site-specific research protocols for future clinical trials on this topic.

## Introduction

Earthquakes cause not only physical impairments but also psychological stresses among their victims and are one of the natural catastrophes with the greatest destructive effect among all natural disasters [1]. Earthquakes have drawn attention for their frequent occurrence and massive destruction over the past several decades [2]. There have been about 1,300 more than about 2,500 earthquakes with a magnitude of 5.0 or higher every year worldwide [3].

Some survivors suffer severe trauma-related symptoms since natural disasters such as earthquakes are unpredictable and destructive [4]. Disaster survivors with high exposure levels to earthquakes have a higher rate of major psychiatric disorders (MPD), such as posttraumatic stress disorder (PTSD), major depressive disorder (MDD), other anxiety disorders (AD), and nicotine dependence, that is 1.4 times higher than of those survivors who are not exposed [4]. PTSD is a common mental health problem among earthquake survivors according to recent reviews [5]. Symptomatic PTSD and depression can persist among many survivors, even 8 years later [6]. PTSD occurs after traumatic events such as death, serious injury, disaster, or sexual violence. Patients experience intrusion symptoms, persistent avoidance of stimuli, negative alteration in cognition, marked alteration in arousal, and angry outbursts. It also causes severe social and psychological disabilities [7]. In particular, when PTSD occurs due to earthquake exposure, 4.10%–67.07% of adults and 2.50%–60.00% of children experience PTSD [8]. PTSD among earthquake survivors can cause comorbid behavioral disorders, substance abuse, self-harm, depression, anxiety, and even suicidal thoughts or impulses [9, 10].

One of the most common MPDs following a natural disaster in survivors is also MDD [11]. The prevalence of MDD (28.3%) following the Haiti earthquake exposure was similar to that of PTSD (24.6%) [12]. The prevalence of MDD and depressive symptoms varied from 1.1% to 30%, and depressive symptoms still persisted even after 2 years according to the study of the prevalence of tsunami-related MDD in Thai survivors [13]. Moreover, MDD symptoms increased with time among firefighters following natural disasters [14]. In particular, since MDD is associated with suicide, evaluation and management of MDD are very important to care for mental health in disaster survivors [15, 16]. Information on AD prevalence following disasters and PTSD is still insufficient. However, it is possible to understand that high distress and disorganization are the disaster triggers for AD [17]. In a wildfire in 2018, the possibility of general AD symptom development could be up to 7 times in survivors who have pre-existing AD, watched homes being destroyed, living in a different place, and receiving inadequate support or counseling [18]. In the Jiuzhaigou earthquake in 2017, research showed that the prevalence of PTSD was 52.7%, 53.8% for anxiety symptoms, and 69.6% for depressive symptoms in the severely affected area [19]. MPD prevalence was high even in the first year after the earthquake exposure [20].

Currently, MPD treatments following disasters are typically pharmacological and psychological treatments [21–23] Selective serotonin reuptake inhibitor is a typical pharmacological treatment for MPD. However, constipation, diarrhea, dizziness, nausea, and sexual dysfunction are the adverse effects [22, 24, 25]. Benzodiazepine can cause drowsiness, falls, or overdoses and even carry significant medical complications, including delirium, although it is also used for MPD [26] Psychological treatments, including prolonged exposure, cognitive

processing therapy, cognitive behavior therapy (CBT), and eye movement desensitization therapy, are strongly recommended treatments by clinical practice guidelines for the management of PTSD [27, 28] A recent meta-analysis showed robust evidence that CBT with a trauma focus (CBT-T), as well as Eye Movement Desensitization and Reprocessing, had a clinically important effect, and prolonged exposure, cognitive processing therapy, and CBT had the strongest evidence of effect [29]. However, many of the patients fail to complete the treatment course because CBT-T also stimulates traumatic recalls [30] Psychological treatment requires an expert practitioner and is not always feasible with long treatment waiting lists resulting from a limited number of qualified therapists [31] CBT-T, for example, is not covered by the National Health Services or health insurance in all countries, thereby excluding individuals, in such countries, who cannot afford such treatment [32]. Internet-delivered CBT-T may not be suitable for survivors who are not comfortable with technology [33]. Survivors cannot access current devices or Internet services when electricity supply and communications systems are shut off following an earthquake.

Acupuncture is a typical complementary integrative medicine modality that is growing as an alternative therapy for MPD [34]. A number of studies on the effectiveness and safety of acupuncture as a treatment for PTSD, MDD, and AD have been published [35–37]. Acupuncture therapy has been used in PTSD, MDD, and AD among earthquake survivors in previous studies [38, 39]. Acupuncture is a medical tool that can be used immediately in a disaster setting, has scarce medical resources, and is effective for not only physical but also psychological symptoms [40, 41]. PTSD is often accompanied by chronic pain [42], and acupuncture has a significant advantage in easing pain [43]. Acupuncture therapy is an easier and cheaper treatment than CBT-T or pharmacological treatment [44]. It is also safe, with only six cases with side effects among the 760,000 treated with acupuncture [45]. However, acupuncture does not lead patients to drop out as the treatment does not provoke trauma recall [41]. Acupuncture treatment has an anti-inflammatory effect and affects the HPA axis or the autonomic nervous system. It has shown effects on disorders such as depression and anxiety that may occur with PTSD [46]. It activates neurotransmitters and have effect on depression, anxiety, and PTSD, respectively [47]. Various acupuncture treatments such as electroacupuncture and pharmacopuncture have also been used for Wenchuan earthquake survivors in addition to simple acupuncture [48].

When exploring a wide range of questions, for example, what kind of research design was used, how are the concepts and characteristics of existing literature, or identifying knowledge gaps, scoping review is more appropriate than a systematic review [49]. The scoping review method is actively used in researches on psychological intervention for disaster [50–52]. Our research team was determined to carry out a scoping review that has a wider view of the relevant field than a systematic review of randomized controlled studies since acupuncture research on PTSD of earthquake survivors has not yet been actively conducted. A previous systematic review was conducted to summarize clinical studies using ear acupuncture for psychological trauma-related disorders after large-scale disasters [53]. However, this review included studies on not specific disasters such as earthquakes but large-scale disasters and included participants with only PTSD and known PTSD-related symptoms. To the best of our knowledge, no scoping review of clinical studies using acupuncture for MPD following exposure to earthquakes has been conducted. Our scoping review aimed to identify which type of clinical research design had been administered on acupuncture treatment for MPD in earthquake survivors. We will also focus on detailed methodological issues such as treatment regimen, detailed characteristics of participants, and frequently used outcomes. We can examine the research gap between the clinical studies and the demands of physicians regarding clinical evidence. Our results will provide fundamental data in choosing the appropriate research

questions and designs when preparing further clinical research and systematic review. No study has shown the status of acupuncture studies on MPD following earthquake exposure, although there are various acupuncture studies for MPD. This study aims to present the current status of acupuncture in MPD following earthquakes and propose acupuncture therapy as an alternative to conventional treatments.

## Materials and methods

### Study design and registration

The scoping review will be conducted according to the method used by Arksey, O'Malley [54], and others [55, 56]. Additionally, it was aligned with PRISMA Extension for Scoping Reviews (PRISMA-Scr). On the other hand, the protocol was submitted to the Open Science Framework (https://osf.io/wfru7/) on July 23, 2022 [50].

### Stage 1: Identifying the study questions

In this stage, the research team will agree to navigate the available clinical evidence in the literature to generate the best scoping research questions according to the consensus between team members. Team members will include specialists in neuropsychiatry (SHK, CYK, DWK), a specialist in clinical research on traditional East Asian medicine (JTL), and an undergraduate researcher (HJK). After the agreement on the revisions and the review concepts, the resulted research questions were as follows:

Regarding survivors following an earthquake,

1. Which clinical research designs were used in previous investigations on the use of acupuncture to treat MPD?

2. What is the frequently used acupuncture type for MPD management?

3. Which clinical outcomes were adopted in previous studies on MPD management?

4. What kind of adverse events happened after acupuncture therapy for MPD?

5. How long should acupuncture treatment be administered for MPD management?

6. Which populations were the target in previous MPD acupuncture studies?

### Stage 2: Identifying relevant studies

**Information source.** Out of many MPDs, this review will focus on PTSD, MDD, and AD (including related symptoms). A literature search will be conducted from inception to November 2022. However, the included databases are Medline (via PubMed), Excerpta Medica data-BASE, Cochrane Central Register of Controlled Trials, Web of Science, Scopus, Allied and Complementary Medicine Database, Cumulative Index to Nursing and Allied Health Literature, PsycArticles, China National Knowledge Infrastructure, Wanfang, VIP, Oriental Medicine Advanced Searching Integrated System, Korea Citation Index, and Citation Information by NII. To avoid bias and include all relevant papers, gray literature will be searched using google scholar, reference lists of relevant systemic reviews or retrieved articles will be searched manually, and the authors of any inaccessible published paper will be contacted. Our research team conducted several literature reviews on acupuncture treatment. Therefore, we applied the existing proven search strategy in our previous acupuncture reviews. Even in the case of search terms related to earthquakes, the search strategy in our previous systematic reviews related to earthquakes was referenced. The previous search strategy was approved after

discussion with a clinical researcher, an expert on literature review, and a specialist on psychiatric diseases. Search terms consisted of disease (earthquake exposure) and intervention terms (acupuncture). Only terms associated with earthquakes will be used in the search phrase, with various synonyms and related medical subject headings. The search terms (S1 Table), inclusion, and exclusion criteria were developed with the consensus of the research team.

**Eligibility criteria: Study types.** Any clinical studies on the effects of acupuncture on MPD patients following an earthquake will be included. The acceptable study designs will be systematic reviews, randomized, quasi-randomized, or nonrandomized controlled clinical trials, single-arm trials, case series, cross-sectional studies, and feasibility studies. On the other hand, case reports with fewer than three patients [57], literature reviews, and preclinical studies will be excluded.

**Eligibility criteria: Types of participants.** PTSD, MDD, and AD (including related symptoms) survivors after the earthquake will be included in addition to MPD patients with musculoskeletal pain. However, we will include studies regardless of using standardized diagnostic criteria for MPD, such as the Diagnostic and Statistical Manual of Mental Disorders and International Classification of Diseases.

**Eligibility criteria: Intervention types.** Various acupuncture therapies will be included such as manual acupuncture, electroacupuncture, bee-venom acupuncture, pharmacopuncture, warm-needle acupuncture, fire needle acupuncture, and acupotomy. On the other hand, acupressure therapy that did not insert needles at acupoints will not be included. The treatment period, dosage, treatment frequency, and concomitant treatment will not be restricted. Except for East Asian traditional medicine interventions, such as herbal medicine, moxibustion, cupping, and tui-na, any type of control group intervention will be included. Furthermore, we also allow major psychological treatments regarding PTSD as a control group intervention (S2 Table).

**Eligibility criteria: Outcome measurements.** All symptoms after diagnosis of MPD will be considered in addition to adverse events and dropout rates related to treatment. The PTSD outcomes will be categorized according to a previous study [58], as follows:1) outcomes related to psychological aspects (anxiety, fear, anger, irritability, guilt, shame, apathy, distrust, sadness, frustration, alienation, loss of confidence, and mourning); 2) outcomes related to somatic aspects (insomnia, palpitation, pain, anorexia, and fatigue); 3) outcomes related to cognitive aspects (decreased memory, difficulty making decisions, repeated recall of traumatic events, and difficulty concentrating). In terms of safety, we will also investigate the incidence of adverse events and dropout rates.

## Stage 3: Study selection

Two reviewers (HJK and DWK) will independently eliminate duplicate publications, then screen the titles and abstracts of the articles for inclusion criteria. In the second phase, the full text will also be screened for the inclusion criteria. However, the inclusion and exclusion reasons will be recorded for each article. In cases of discrepancies, the independent researcher will intervene to solve the disagreement. Details of the study selection process are shown in Fig 1.

## Stage 4: Charting the data

A data extraction sheet was prepared after the study team's pilot test. The items of the extraction sheet included: 1) general information, such as the first author's name, country, publication year, and research design; 2) participant demographic data, such as age, sex, number of participants (initial and final), diagnostic criteria, and disease duration and severity; 3) detailed

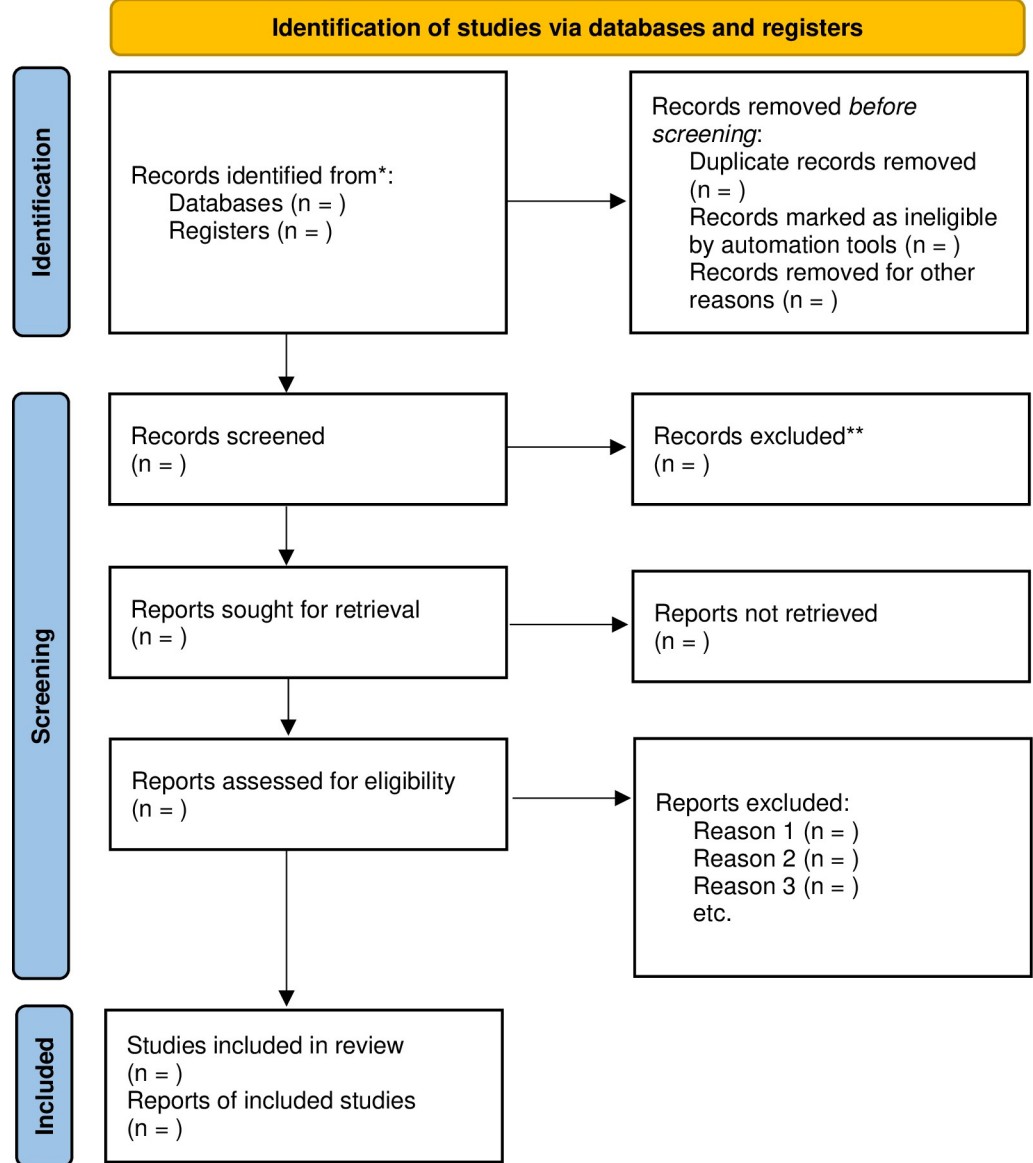

**Fig 1. PRISMA flowchart for the study selection process.**

information on interventions such as type of acupuncture, acupuncture points, treatment dosage (number, frequency), treatment period, and details of control/concomitant interventions; and 4) outcome variables. On the other hand, effects, safety, and research findings data will also be extracted.

Two reviewers (HJK and DWK) will independently conduct the extraction and will crosscheck the data from all included articles. Any disagreement will be solved after discussion with a third independent researcher (SHK).

## Stage 5: Collating, summarizing, and reporting the results

The extracted data will be collated, synthesized, and summarized according to the analytical framework of the scoping review.

In the qualitative analysis stage, we will describe the characteristics of included studies, including the author's name, country, publication year, earthquake details, number of participants, sex, age, research design, and type of treatment/control group interventions. The second table titled "Detailed information of acupuncture treatment" will include the characteristics of the intervention such as acupuncture type, location of acupuncture points, number of treatments, treatment frequency, treatment period, and details of control/concomitant interventions. Finally, the third table titled "Effects and safety of acupuncture treatment for MPD and related symptoms" will include the outcomes, adverse events, and conclusions of each study. Additionally, a table titled "Research map" will be provided to help with future research planning by visualizing the state of ongoing research and commonly utilized findings. The results of this study may be used to identify knowledge gaps in the body of research on acupuncture therapy for MPD.

## Ethics and dissemination

Ethics approval and informed consent are not required because this study will be based exclusively on published literature, in which informed consent was obtained by the primary investigators. We will publish this scoping review in a peer-reviewed journal.

## Discussion

The current treatment has some limitations although MPD occurrence in earthquake survivors is an important problem. Acupuncture therapy is an alternative to conventional treatment [41, 48]. There is currently no comprehensive acupuncture therapy review for MPD following an earthquake. Therefore, we will perform the first scoping review.

We will identify possible studies and research methods in this field through this study. We will also summarize acupuncture regimen, subject characteristics, outcomes, and types of study designs. We will clarify knowledge gaps for future studies. This research brings together the existing studies and will help us plan our follow-up research. Therefore, it could reduce the burden of researchers conducting subsequent studies. These data also could provide foundational information for clinicians to manage MPD after an earthquake.

Nevertheless, our study has some limitations. First, the outcome, including adverse reactions, frequency, or duration of acupuncture, may be obscured since there will be no restrictions on concurrent treatment with acupuncture. Depending on the design of the included study, for example, when various concurrent treatments were implemented, these may affect interpretation of the acupuncture outcome. Second, although we will perform a comprehensive research, related articles in languages other than English, Korean, Japanese, and Chinese may be excluded. Third, consultation was not planned, which is the last step of scoping registration and an optional sixth step. This is because it is difficult and unethical to set up artificial mentally disabled patients.

This research only used information from previous articles, so ethical consideration or informed consent is not needed. Research findings will be submitted to an academic journal and conferences for further distribution. Moreover, we will also develop an e-leaflet to provide the key findings of our review via social network services to the research community.

## Conclusions

The findings of this scoping review will provide fundamental data that will help researchers identify appropriate research questions and design further studies on the use of acupuncture for MPD management in earthquake survivors. These results will be helpful for developing disaster site-specific research protocols for future clinical trials on this topic

## Supporting information

**S1 Table. Search terms used in each database.**
(DOCX)

**S2 Table. Central characteristics of existing psychological therapies for PTSD.**
(DOCX)

**S3 Table. PRISMA-P (Preferred Reporting Items for Systematic review and Meta-Analysis Protocols) 2015 checklist: Recommended items to address in a systematic review protocol.**
(DOC)

## Author Contributions

**Conceptualization:** Jungtae Leem, Sang-Ho Kim.

**Data curation:** Hui-Ju Kwon, Da-Woon Kim.

**Funding acquisition:** Sang-Ho Kim.

**Investigation:** Hui-Ju Kwon, Da-Woon Kim.

**Methodology:** Jungtae Leem, Chan-Young Kwon.

**Software:** Da-Woon Kim, Sang-Ho Kim.

**Writing – original draft:** Hui-Ju Kwon.

**Writing – review & editing:** Chan-Young Kwon, Sang-Ho Kim.

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
