## [Decision Letter · Decision Letter 0]

8 Nov 2022

PONE-D-22-25388Effect of acupuncture on patients with major psychiatric disorder and related symptoms caused by earthquake: Protocol for a scoping review of clinical studiesPLOS ONE

Dear Dr Sang-Ho Kim,

Thank you for submitting your manuscript to PLOS ONE. After careful consideration, we feel that it has merit but does not fully meet PLOS ONE’s publication criteria as it currently stands. Therefore, we invite you to submit a revised version of the manuscript that addresses the points raised during the review process.

It is a very interesting and pertinent topic to consider. Please consider the contributions and comments of the reviewers, which will clarify and enrich the quality of the manuscript.

We look forward to receiving your revised manuscript.

Kind regards,

Juan-Luis Castillo-Navarrete, Ph.D.

Academic Editor

PLOS ONE

Journal Requirements:

3.  Please review your reference list to ensure that it is complete and correct. If you have cited papers that have been retracted, please include the rationale for doing so in the manuscript text or remove these references and replace them with relevant current references. Any changes to the reference list should be mentioned in the rebuttal letter that accompanies your revised manuscript. If you need to cite a retracted article, indicate the article’s retracted status in the References list and also include a citation and full reference for the retraction notice.

Reviewers' comments:

Reviewer's Responses to Questions

**Comments to the Author**

1. Does the manuscript provide a valid rationale for the proposed study, with clearly identified and justified research questions?

Reviewer #1: Yes

Reviewer #2: Yes

2. Is the protocol technically sound and planned in a manner that will lead to a meaningful outcome and allow testing the stated hypotheses?

Reviewer #1: Yes

Reviewer #2: Partly

3. Is the methodology feasible and described in sufficient detail to allow the work to be replicable?

Reviewer #1: Yes

Reviewer #2: Yes

4. Have the authors described where all data underlying the findings will be made available when the study is complete?

Reviewer #1: Yes

Reviewer #2: Yes

5. Is the manuscript presented in an intelligible fashion and written in standard English?

Reviewer #1: Yes

Reviewer #2: Yes

6. Review Comments to the Author

You may also provide optional suggestions and comments to authors that they might find helpful in planning their study.

Reviewer #1: Recommendations for authors:

In the Title.

It is suggested to specify that the cause of the disorders is caused by "exposure" to an earthquake.

In the abstract:

It is suggested to moderate the phrase "earthquakes have the greatest destructive effect on natural catastrophes". Perhaps the authors could point out that earthquakes are one of the natural catastrophes with the greatest destructive effect.

Another aspect of wording to improve in the abstract is similar to the problem with the title. The sentence that reads: Posttraumatic stress disorder (PTSD), major depressive disorder (MDD), and anxiety disorder (AD) are major psychiatric disorder (MPD) that can be triggered by earthquakes. … This sentence could be modified to read: "that can be triggered in people by exposure to an earthquake...".

In the results and conclusions, although these sections are projection sections, they are very generic, and applicable to any study. They are almost like model sentences. Authors are required to write projection results and projection conclusions specific to the topic of interest and the target audience. These should generate an impact on the reader regarding what is expected to be found and the usefulness of these findings.

In the introduction.

On line 108, the authors state: “…Psychological treatments include eye movement desensitization therapy, prolonged exposure, ognitive processing therapy, and cognitive behavior therapy…”. The authors could include a table with the types of existing therapies and include their central characteristics, for example: authors, what they consist of, history of effectiveness, benefits, limitations. This would allow the reader to easily observe and compare the types of therapy.

It would be important for the authors to check if there are previous systematic reviews or scoping reviews. It is important to know whether or not there is a systematic review or scoping review on the topic of study. Perhaps there are similar reviews, in this sense it is relevant to know the differences and contribution of the new proposed review in relation to the existing ones on the same or similar research topic. For example, analyze the proposed objectives, databases consulted, limitations of existing reviews, etc..

For example, a quick exploration in Google scholar shows the following related systematic reviews.

• Kwon, C. Y., Lee, B., & Kim, S. H. (2020). Effectiveness and safety of ear acupuncture for trauma-related mental disorders after large-scale disasters: A PRISMA-compliant systematic review. Medicine, 99(8).

• Moiraghi, C., Poli, P., & Piscitelli, A. (2019). An observational study on acupuncture for earthquake-related post-traumatic stress disorder: the experience of the lombard association of medical acupuncturists/acupuncture in the world, in Amatrice, Central Italy. Medical acupuncture, 31(2), 116-122.

• Hong, C., & Efferth, T. (2016). Systematic review on post-traumatic stress disorder among survivors of the Wenchuan earthquake. Trauma, Violence, & Abuse, 17(5), 542-561.

• Ding, N., Li, L., Song, K., Huang, A., & Zhang, H. (2020). Efficacy and safety of acupuncture in treating post-traumatic stress disorder: A protocol for systematic review and meta-analysis. Medicine, 99(26).

It is also possible to identify scoping reviews on the topic, as the authors point out that this type of review is more relevant in coherence with the proposed research questions (line 137 to 140.): “Scoping review is more appropriate than a systematic review.(46) Our research team was determined to carry out a scoping review which has a wider view of the relevant field than a systematic review of randomized controlled studies since acupuncture research on PTSD of earthquake survivors has not yet been actively conducted”.

• Zahos, H., Crilly, J., & Ranse, J. (2022). Psychosocial problems and support for disaster medical assistance team members in the preparedness, response and recovery phases of natural hazards resulting in disasters: A scoping review. Australasian emergency care.

• Nascimento, J., Santos, K., Dantas, J., Dantas, D., & Dantas, R. (2021). Non-pharmacological therapies for the treatment of post-traumatic stress disorder among emergency responders: a scoping review. Revista da Escola de Enfermagem da USP, 55.

• Kim, J., Chesworth, B., Franchino-Olsen, H., & Macy, R. (2021). A scoping review of vicarious trauma interventions for service providers working with people who have experienced traumatic events. Trauma, Violence, & Abuse, 1524838021991310.

In addition, it is considered essential to review previous studies that have used both review approaches (scoping reviews and systematic reviews) since the authors state in the protocol method that they will use PRISMA: línea 155 y 156 “ Additionally, it was aligned with PRISMA and its Extension for Scoping Reviews (PRISMA-Scr)”.

Method

In line 178 the authors point out that: “.. A literature search will be conducted from inception to June 2022…”. However, we are already in October 2022. It would be important to specify that the search will be updated.

In the search strategy it is important to mention the validation procedure of the search algorithm. It would also be important to consider previous systematic reviews on the subject and to give an account of their search algorithms. This is a procedure that allows the authors to support the selected keywords to build their own search algorithm which is also required to be validated by experts in the field.

Considering that the authors point out within their objectives methodological aspects of the research, as well as the results of cynical studies that have implemented acupuncture (L 140 to L 145), it would be interesting for them to consider the incorporation of the measurement methods used in the studies.

Reviewer #2: The justification for the study is adequately presented and the methodology is clear. However, the following issues must be resolved for the article to be eligible for publication:

Lines 74-102: The information presented is valuable, but should be better organized.

Lines 103-116: Several psychological and pharmacological treatments are presented without specific indication for certain disorders for each one. In addition, only their secondary negative effects are described without describing their primary effect, the therapeutic benefit. Please, provide effect sizes for therapeutic effects of each treatment.

“Acupuncture was an immediate medical tool in the meantime and was effective for not only physical but also psychological symptoms, but has scarce medical resources”. Check redaction

“CBT also stimulates traumatic recalls.” This should be included on previous paragraph.

Line 136: The interrogation sign was a typo? Check

Line 177: Web of Science and Scopus are databases used in many reviews. Please, include them or provide rationality for not include them.

Line 178 – vs 189: “Out of many MPDs, this review will focus on PTSD” vs “The kind of MPD will not be restricted”. This is contradictory. Please, resolve.

Line 195-199: “The acceptable study designs will be reviews” vs “On the other hand, […] literature reviews […] will be excluded”. This is contradictory. Please, resolve.

“On the other hand, acupressure therapy will not be included”. Justify this decision.

“Except for East Asian traditional medicine interventions, such as herbal medicine, moxibustion, cupping, and tui-na, any type of control group intervention will be included”. Please, justify this decision.

7. PLOS authors have the option to publish the peer review history of their article (what does this mean?). If published, this will include your full peer review and any attached files.

Reviewer #1: **Yes: **Fabiola Sáez-Delgado

Reviewer #2: **Yes: **Claudio Bustos Navarrete

---

## [Author Response · Author response to Decision Letter 0]

29 Nov 2022

Response to the Reviewers’ Comments

Title: Effect of acupuncture on patients with major psychiatric disorder and related symptoms caused by earthquake exposure: Protocol for a scoping review of clinical studies

Emily Chenette 

Editor-in-Chief

PLOS ONE

Dear Editor Juan-Luis Castillo-Navarrete, Ph.D:

I wish to re-submit the manuscript titled “Effect of acupuncture on patients with major psychiatric disorder and related symptoms caused by earthquake exposure: Protocol for a scoping review of clinical studies.” The manuscript ID is PONE-D-22-25388.

We thank you and the reviewers for your thoughtful suggestions and insights. The manuscript has benefited from these insightful suggestions. I look forward to working with you and the reviewers to move this manuscript closer to publication in the PLOS ONE.

The manuscript has been rechecked and the necessary changes have been made in accordance with the reviewers’ suggestions. The responses to all comments have been prepared and attached herewith/given below. 

Thank you for your consideration. I look forward to hearing from you.

Response to Journal Requirements 

Comment #1: 

Response #1:

Thank you for your comment. I have reviewed manuscript carefully to meet style requirements. I changed some parts (marked in yellow).

I delete keywords. And I moved Ethics and dissemination to methods section. 

“The search terms and strategies are detailed in S1 Table.”

“Conclusions”

“Supporting information

S1 Table. Search terms used in each database.

S2 Table. Central characteristics of existing psychological therapies for PTSD

S3 Table. PRISMA-P (Preferred Reporting Items for Systematic review and Meta-Analysis Protocols) 2015 checklist: recommended items to address in a systematic review protocol*

Comment #2: 

We note that the grant information you provided in the ‘Funding Information’ and ‘Financial Disclosure’ sections do not match. 

Response #2:

When you resubmit, I did match the grant information in the ‘Funding Information’ and ‘Financial Disclosure’ sections.

“This work was supported by the National Research Foundation of Korea (NRF) grant funded by the Korean government (MSIT) (No. 2021R1F1A105928211). SHK has received this fund. The funding source had no input in the interpretation or publication of the study results. The funders had no role in study design, data collection and analysis, decision to publish, or preparation of the manuscript. ※ MSIT: Ministry of Science and ICT”

Comment #3: 

Please review your reference list to ensure that it is complete and correct. If you have cited papers that have been retracted, please include the rationale for doing so in the manuscript text or remove these references and replace them with relevant current references. Any changes to the reference list should be mentioned in the rebuttal letter that accompanies your revised manuscript. If you need to cite a retracted article, indicate the article’s retracted status in the References list and also include a citation and full reference for the retraction notice.

Response #3:

I have checked all reference lists. There are no retracted papers. I also have corrected information of some references such as issue and page. 

9. Keane TM, Brief DJ, Pratt EM, Miller MW. Assessment of PTSD and its comorbidities in adults. Handbook of PTSD: Science and practice. The Guilford Press; 2007:279-305.

17. McFarlane AC, Van Hooff M, Goodhew F. Anxiety disorders and PTSD. Mental health and disasters. Cambridge: Cambridge University Press; 2012:47-66.

22. Qaseem A, Barry MJ, Kansagara D. Clinical Guidelines Committee of the American College of Physicians. Nonpharmacologic versus pharmacologic treatment of adult patients with major depressive disorder: a clinical practice guideline from the American College of Physicians. Annals of internal medicine. 2016;164(5):350-9.

40. Takayama S, Kamiya T, Watanabe M, Hirano A, Matsuda A, Monma Y, et al. Report on disaster medical operations with acupuncture/massage therapy after the great East Japan earthquake. Integrative Medicine Insights. 2012;7. S9541.

58. Ursano RJ, Fullerton CS, Weisaeth L, Raphael B. Textbook of disaster psychiatry: Cambridge: Cambridge University Press; 2017.

Response to Comments from Reviewer 1

In the Title

Comment #1: 

It is suggested to specify that the cause of the disorders is caused by "exposure" to an earthquake. 

Another aspect of wording to improve in the abstract is similar to the problem with the title. The sentence that reads: Posttraumatic stress disorder (PTSD), major depressive disorder (MDD), and anxiety disorder (AD) are major psychiatric disorder (MPD) that can be triggered by earthquakes. … This sentence could be modified to read: "that can be triggered in people by exposure to an earthquake...". 

Response #1:

Thank you for your comment. According to your suggestion, I have changed the title and the sentence in the abstract and introduction.

“Effect of acupuncture on patients with major psychiatric disorder and related symptoms caused by earthquake exposure: Protocol for a scoping review of clinical studies“

“Posttraumatic stress disorder (PTSD), major depressive disorder (MDD), and anxiety disorder (AD) are major psychiatric disorder (MPD) that can be triggered by exposure to earthquakes” [Line 84, page 4]

“…In particular, when PTSD occurs due to an earthquake exposure,…” [Line 84, page 4]

“.. The prevalence of MDD (28.3%) following the Haiti earthquake exposure…”[Line 88, page 4]

“MPD prevalence was high even in the first year after the earthquake exposure.(20)” [Line 102, page 5]

“has shown the status of acupuncture studies on MPD following earthquake exposure, although there are” [Line 152, page 7]

In the abstract:

Comment #2: 

It is suggested to moderate the phrase "earthquakes have the greatest destructive effect on natural catastrophes". Perhaps the authors could point out that earthquakes are one of the natural catastrophes with the greatest destructive effect.

Response #2:

According to your suggestion, I have changed the phrase in the abstract and introduction.

“earthquakes are one of the natural catastrophes with the greatest destructive effect.”

Comment #3: 

In the results and conclusions, although these sections are projection sections, they are very generic, and applicable to any study. They are almost like model sentences. Authors are required to write projection results and projection conclusions specific to the topic of interest and the target audience. These should generate an impact on the reader regarding what is expected to be found and the usefulness of these findings. 

Response #3:

Thank you for insightful comment. According to your suggestion, I did re-write the conclusion.

“The findings of this scoping review will provide fundamental data that will help researchers identify appropriate research questions and design further studies on the use of acupuncture for MPD management in earthquake survivors. These results will be helpful for developing disaster site-specific research protocols for future clinical trials on this topic. (marked in yellow).” [page 3, 13] 

In the introduction.

Comment #4: 

On line 108, the authors state: “…Psychological treatments include eye movement desensitization therapy, prolonged exposure, cognitive processing therapy, and cognitive behavior therapy…”. The authors could include a table with the types of existing therapies and include their central characteristics, for example: authors, what they consist of, history of effectiveness, benefits, limitations. This would allow the reader to easily observe and compare the types of therapy.

Response #4:

Thank you for comment. According to your suggestion, I added table regarding central characteristics of existing psychological therapies for PTSD (S2 table). 

Comment #5: 

It would be important for the authors to check if there are previous systematic reviews or scoping reviews. It is important to know whether or not there is a systematic review or scoping review on the topic of study. Perhaps there are similar reviews, in this sense it is relevant to know the differences and contribution of the new proposed review in relation to the existing ones on the same or similar research topic. For example, analyze the proposed objectives, databases consulted, limitations of existing reviews, etc.. 

For example, a quick exploration in Google scholar shows the following related systematic reviews.

• Kwon, C. Y., Lee, B., & Kim, S. H. (2020). Effectiveness and safety of ear acupuncture for trauma-related mental disorders after large-scale disasters: A PRISMA-compliant systematic review. Medicine, 99(8).

• Moiraghi, C., Poli, P., & Piscitelli, A. (2019). An observational study on acupuncture for earthquake-related post-traumatic stress disorder: the experience of the lombard association of medical acupuncturists/acupuncture in the world, in Amatrice, Central Italy. Medical acupuncture, 31(2), 116-122.

• Hong, C., & Efferth, T. (2016). Systematic review on post-traumatic stress disorder among survivors of the Wenchuan earthquake. Trauma, Violence, & Abuse, 17(5), 542-561.

• Ding, N., Li, L., Song, K., Huang, A., & Zhang, H. (2020). Efficacy and safety of acupuncture in treating post-traumatic stress disorder: A protocol for systematic review and meta-analysis. Medicine, 99(26).

It is also possible to identify scoping reviews on the topic, as the authors point out that this type of review is more relevant in coherence with the proposed research questions (line 137 to 140.): “Scoping review is more appropriate than a systematic review.(46) Our research team was determined to carry out a scoping review which has a wider view of the relevant field than a systematic review of randomized controlled studies since acupuncture research on PTSD of earthquake survivors has not yet been actively conducted”.

• Zahos, H., Crilly, J., & Ranse, J. (2022). Psychosocial problems and support for disaster medical assistance team members in the preparedness, response and recovery phases of natural hazards resulting in disasters: A scoping review. Australasian emergency care.

• Nascimento, J., Santos, K., Dantas, J., Dantas, D., & Dantas, R. (2021). Non-pharmacological therapies for the treatment of post-traumatic stress disorder among emergency responders: a scoping review. Revista da Escola de Enfermagem da USP, 55.

• Kim, J., Chesworth, B., Franchino-Olsen, H., & Macy, R. (2021). A scoping review of vicarious trauma interventions for service providers working with people who have experienced traumatic events. Trauma, Violence, & Abuse, 1524838021991310.

Response #5:

Thank you so much for kind and insightful comments. According to your suggestion, I added one similar review and the differences of our scoping review. We already have cited Moiraghi,(2019) in the introduction section. Hong (2016)’s study is not for clinical studies using acupuncture. Ding(2020)’s study is not for disaster. Up to date, there is no scoping review regarding to clinical studies using acupuncture for MPD following exposure to an earthquakes

 “A previous systematic review was conducted to summarize clinical studies using ear acupuncture for psychological trauma-related disorders after large-scale disasters.53 However, this review included studies on not specific disasters such as earthquakes but large-scale disasters and included participants with only PTSD and known PTSD-related symptoms. To the best of our knowledge, no scoping review of clinical studies using acupuncture for MPD following exposure to earthquakes has been conducted.”[page 7]

Method

Comment #6: 

In addition, it is considered essential to review previous studies that have used both review approaches (scoping reviews and systematic reviews) since the authors state in the protocol method that they will use PRISMA: línea 155 y 156 “ Additionally, it was aligned with PRISMA and its Extension for Scoping Reviews (PRISMA-Scr)”.

Response #6:

Our protocol is a scoping review. We changed sentence in the method to clarify the meaning. And we added recent studies to review previous studies that have used PRISMA-Scr. 

The scoping review method is actively used in researches on psychological intervention for disaster.50-52 

“PRISMA and its Extension for Scoping Reviews => PRISMA Extension for Scoping Reviews”

Comment #7: 

In line 178 the authors point out that: “.. A literature search will be conducted from inception to June 2022…”. However, we are already in October 2022. It would be important to specify that the search will be updated.

Response #7:

We will update the search. 

“A literature search will be conducted from inception to November 2022.”

Comment #8: 

In the search strategy it is important to mention the validation procedure of the search algorithm. It would also be important to consider previous systematic reviews on the subject and to give an account of their search algorithms. This is a procedure that allows the authors to support the selected keywords to build their own search algorithm which is also required to be validated by experts in the field.

Response #8:

We added the validation procedure of the search algorithm to consider previous systematic reviews.

“Our research team conducted several literature review about acupuncture treatment. Therefore, we applied the existing proven search strategy in previous acupuncture reviews. Even in the case of search terms related to earthquakes, the search strategy in previous SRs related to earthquakes was referenced.(page 9)”

Comment #9: 

Considering that the authors point out within their objectives methodological aspects of the research, as well as the results of cynical studies that have implemented acupuncture (L 140 to L 145), it would be interesting for them to consider the incorporation of the measurement methods used in the studies.

Response #9:

Thank you for the good point. From the beginning of the study, we tried to minimize bias by including both subjective and objective variables in the selection of outcome variables.

Response to Comments from Reviewer 2

Comment #1: 

Lines 74-102: The information presented is valuable, but should be better organized.

Response #1:

Thank you for comment. We have separated that paragraphs for organization and readability. 

“……. substance abuse, self-harm, depression, anxiety, and even suicidal thoughts or impulses.(9, 10)

 One of the most common MPDs following a natural disaster in survivors is also MDD.(11) The……..”

Comment #2: 

Lines 103-116: Several psychological and pharmacological treatments are presented without specific indication for certain disorders for each one. In addition, only their secondary negative effects are described without describing their primary effect, the therapeutic benefit. Please, provide effect sizes for therapeutic effects of each treatment.

Response #2:

Thank you for insightful comment. I changed relevant sentence. And I added sentence regarding effect sizes for therapeutic effects of each treatment using recent meta-analysis.

“Psychological treatments, including prolonged exposure, cognitive processing therapy, cognitive behavior therapy (CBT), and eye movement desensitization therapy, are strongly recommended treatments by clinical practice guidelines for the management of PTSD.27,28 A recent meta-analysis showed robust evidence that CBT with a trauma focus (CBT-T), as well as Eye Movement Desensitization and Reprocessing, had a clinically important effect, and prolonged exposure, cognitive processing therapy, and CBT had the strongest evidence of effect.29 

Comment #3: 

“Acupuncture was an immediate medical tool in the meantime and was effective for not only physical but also psychological symptoms, but has scarce medical resources”. Check redaction

Response #3: 

“Acupuncture was a medical tool that can be utilized immediately in the disaster setting has scarce medical resources and was effective for not only physical but also psychological symptoms.” 

Comment #4: 

“CBT also stimulates traumatic recalls.” This should be included on previous paragraph.

Response #4: 

We moved that sentences to previous paragraph.

“However, many of the patients fail to complete the treatment course because CBT-T also stimulates traumatic recalls.(30)”

Comment #5: 

Line 136: The interrogation sign was a typo? Check

Response #5: 

I deleted the interrogation sign.

“…how are the concepts and characteristics of existing literature, or identifying knowledge gaps,”

Comment #6: 

Line 177: Web of Science and Scopus are databases used in many reviews. Please, include them or provide rationality for not include them.

Response #6: 

According to your suggestion, I added Wos and Scopus in the searching database of abstract and method section.

“the included databases are Medline (via PubMed), Excerpta Medica dataBASE, Cochrane Central Register of Controlled Trials, Web of Science, Scopus, Allied and Complementary Medicine Database, Cumulative Index to Nursing and Allied Health Literature, PsycArticles, China National Knowledge Infrastructure, Wanfang, VIP, Oriental Medicine Advanced Searching Integrated System, Korea Citation Index, and Citation Information by NII.” [page 9]

Comment #7: 

Line 178 – vs 189: “Out of many MPDs, this review will focus on PTSD” vs “The kind of MPD will not be restricted”. This is contradictory. Please, resolve.

Response #7: 

We added MDD, and AD (including related symptoms). And we deleted contradictory sentence.

“Out of many MPDs, this review will focus on PTSD, MDD, and AD (including related symptoms).”[page 9]

The kind of MPD will not be restricted” => delete

Comment #8: 

Line 195-199: “The acceptable study designs will be reviews” vs “On the other hand, […] literature reviews […] will be excluded”. This is contradictory. Please, resolve.

Response #8: 

We added systematic reviews in that sentence.

“The acceptable study designs will be systematic reviews”

Comment #9: 

“On the other hand, acupressure therapy will not be included”. Justify this decision.

Response #9: 

We will only include not acupressure did not insert needles but acupuncuture insert needles through skin at acupoints. 

“On the other hand, acupressure therapy that did not insert needles at acupoints will not be included.”

Comment #10: 

“Except for East Asian traditional medicine interventions, such as herbal medicine, moxibustion, cupping, and tui-na, any type of control group intervention will be included”. Please, justify this decision.

Response #10: 

We want to compare the effectiveness of acupuncture for MPD management in earthquake survivors with usual conventional therapy (medication or psychotherapy) or sham-acupuncture or wait-list group. Therefore, we have excluded East Asian traditional medicine interventions such as herbal medicine, moxibustion, cupping, and tui-na.

---

## [Decision Letter · Decision Letter 1]

18 Jan 2023

Effect of acupuncture on patients with major psychiatric disorder and related symptoms caused by earthquake exposure: Protocol for a scoping review of clinical studies

PONE-D-22-25388R1

Dear Dr. Sang-Ho Kim,

We’re pleased to inform you that your manuscript has been judged scientifically suitable for publication and will be formally accepted for publication once it meets all outstanding technical requirements.

Kind regards,

Juan-Luis Castillo-Navarrete, Ph.D.

Academic Editor

PLOS ONE

Additional Editor Comments (optional):

Reviewers' comments:

Reviewer's Responses to Questions

**Comments to the Author**

1. Does the manuscript provide a valid rationale for the proposed study, with clearly identified and justified research questions?

Reviewer #1: Yes

2. Is the protocol technically sound and planned in a manner that will lead to a meaningful outcome and allow testing the stated hypotheses?

Reviewer #1: Yes

3. Is the methodology feasible and described in sufficient detail to allow the work to be replicable?

Reviewer #1: Yes

4. Have the authors described where all data underlying the findings will be made available when the study is complete?

Reviewer #1: Yes

5. Is the manuscript presented in an intelligible fashion and written in standard English?

Reviewer #1: Yes

6. Review Comments to the Author

You may also provide optional suggestions and comments to authors that they might find helpful in planning their study.

Reviewer #1: The authors responded satisfactorily to the suggestions made. The improved version of the manuscript reflects the improvements incorporated.

7. PLOS authors have the option to publish the peer review history of their article (what does this mean?). If published, this will include your full peer review and any attached files.

Reviewer #1: **Yes: **Fabiola Sáez-Delgado

---

## [Editor Report · Acceptance letter]

20 Jan 2023

PONE-D-22-25388R1 

Effect of acupuncture on patients with major psychiatric disorder and related symptoms caused by earthquake exposure: Protocol for a scoping review of clinical studies 

Dear Dr. Kim:

I'm pleased to inform you that your manuscript has been deemed suitable for publication in PLOS ONE. Congratulations! Your manuscript is now with our production department. 

Kind regards, 

on behalf of

Dr. Juan-Luis Castillo-Navarrete 

Academic Editor

PLOS ONE